# Hydrogel-Based Technologies for the Diagnosis of Skin Pathology

**Christian Wiraja [1,\*], Xiaoyu Ning [1], Mingyue Cui [1] and Chenjie Xu [1,2,3,\*]**

[1]   School of Chemical and Biomedical Engineering, Nanyang Technological University,
     Singapore 637457, Singapore; ning0017@e.ntu.edu.sg (X.N.); mingyue.cui@ntu.edu.sg (M.C.)
[2]   National Dental Centre of Singapore, 5 Second Hospital Ave, Singapore 168938, Singapore
[3]   Department of Biomedical Engineering, City University of Hong Kong, 83 Tat Chee Avenue, Kowloon,
     Hong Kong, China
\*   Correspondence: cwiraja@ntu.edu.sg (C.W.); chenjie.xu@cityu.edu.hk (C.X.)

**Abstract:** Hydrogels, swellable hydrophilic polymer networks fabricated through chemical cross-linking or physical entanglement are increasingly utilized in various biomedical applications over the past few decades. Hydrogel-based microparticles, dressings and microneedle patches have been explored to achieve safe, sustained and on-demand therapeutic purposes toward numerous skin pathologies, through incorporation of stimuli-responsive moieties and therapeutic agents. More recently, these platforms are expanded to fulfill the diagnostic and monitoring role. Herein, the development of hydrogel technology to achieve diagnosis and monitoring of pathological skin conditions are highlighted, with proteins, nucleic acids, metabolites, and reactive species employed as target biomarkers, among others. The scope of this review includes the characteristics of hydrogel materials, its fabrication procedures, examples of diagnostic studies, as well as discussion pertaining clinical translation of hydrogel systems.

**Keywords:** hydrogel; diagnosis; monitoring; skin pathology; skin diseases

## 1. Introduction

Hydrogels are 3D hydrophilic polymer networks that do not dissolve but swell considerably in an aqueous environment. Formed through various assembly mechanisms, namely physical methods (i.e., reversible entanglement and electrostatic interactions) and covalent crosslinking (e.g., amine-carboxylic acid reaction, Schiff base formation, and initiator-mediated in situ polymerization), hydrogels confer unique material properties like tunable mechanical and swelling properties, as well as potentials for on-demand degradation/cargo delivery and self-healing [1,2]. Hydrogels have high water content (70–99%) and excellent biocompatibility, and thus are greatly amiable to tissue applications. Composed of natural (e.g., collagen, alginate) or synthetic polymers (e.g., poly(ethylene glycol)/PEG, poly(acrylic acid)/PAA, and poly(vinyl alcohol)/PVA) with different molecular weights, concentrations, and cross-linking densities, mechanical attributes of hydrogels (i.e., stiffness and toughness) can be adjusted to suit the intended applications [3]. For instance, soft hydrogels are suitable for tissue filling and restoration [4]. Meanwhile, tough and robust hydrogels are necessary for load-bearing applications such as artificial cartilage and muscle [5]. Substituting polymeric components also impacts hydrogel porosity, affecting equilibration of polymer chain interactions with water (i.e., capillary and hydration and osmotic forces) and leading to alteration in the swelling rate and capacity [1]. By employing thermosensitive and/or photolabile linker moieties (e.g., azobenzene and o-nitrobenzyl), hydrogel degradation can be remotely controlled to modulate the release profile of drug cargo molecules [6]. Most recently, the self-healing property of specialized polymer hydrogels is gaining attention for wide

adaptations. Such hydrogels pose the ability to spontaneously form new bonds when old ones are disrupted (i.e., through electrostatic attractions and non-covalent hydrogen bonding), thus promising better retention of hydrogel integrity [7].

Given its facile preparation and versatile, tunable material properties, various hydrogel applications have been researched and explored over the past few decades. These include contact lenses, tissue filler, absorbable cell scaffold for tissue engineering and regenerative medicine, a drug carrier and sensing platform, etc. (Figure 1) [8]. Being injectable and having flexible structures to occupy and close gaps in damaged tissues, hydrogels are proposed as a tissue filler, including but not limited to facial contouring and breast augmentation [9]. In tissue engineering and regenerative medicine, hydrogel scaffolds are widely adopted as 3D templates for culturing stem cells and primary cells. The porous nature of hydrogels facilitates cell migration throughout the scaffold, while permitting transfer and exchange of nutrients and waste metabolites [10–12]. As carriers for drug/sensor moieties, hydrogels promise controllable swelling and bioadhesive properties, thereby enabling versatile loading and tunable, on-demand cargo delivery. For instance, PAA hydrogel with its ability to form hydrogen bonds with the mucosa layer are explored to achieve prolonged oral and vaginal delivery [13,14]. Hydrogel microneedles (MNs) composed of poly(styrene)-block-poly(acrylic acid) tips and non-swellable polystyrene core were shown to achieve 3.5-fold increase in adhesion strength compared to skin stapler, through swelling and mechanical interlocking with nearby tissue [15]. Furthermore, PVA gel MNs facilitate on-demand insulin release following cleavage of labile linkers, based on nearby glucose concentration [16]. From our study, versatile post-assembly loading of hydrogel with hydrophobic and hydrophilic molecules was achieved through the swelling and freeze-spraying method, as an alternative to conventional incorporation preassembly [17,18]. With notable growth in the spectrum of functional mono/polymers and assembly strategies, the applicability of hydrogels is ever expanding as well.

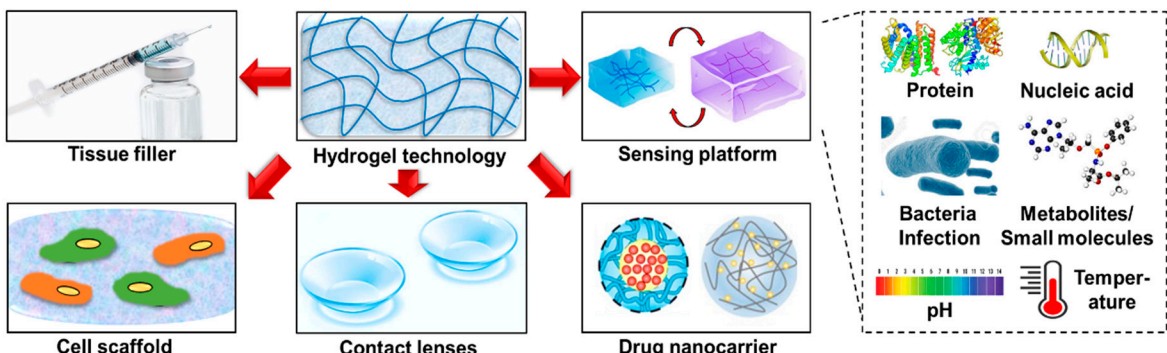

**Figure 1.** Summary of biological applications of hydrogels.

One widely targeted tissue for hydrogel applications is the skin. As the largest organ in the body, the skin is composed of three basic layers including the epidermis, dermis, and the subcutaneous layers. Each layer contains several sublayers [19]. The outermost layer is the avascular epidermis, with a thickness of 100–150 μm, which protects the human body from toxins, bacteria, and fluid loss. The epidermis of thin skin is composed of four distinct sublayers, the stratum basale (SB), stratum spinosum (SS), stratum granulosum (SG), and stratum corneum (SC). The stratum lucidum (SL) is the fifth layer existing between the SC and SG on thicker skin regions of the body. Being the topmost layer of the epidermis, SC serves the majority of the skin barrier function. SC has a thickness of 10–40 μm and is made of 15–20 layers of flattened, dead corneocytes that shed periodically through desquamation [20]. The region beneath the SC is referred to as the viable epidermis, where drug binding, metabolism, surveillance, and active transport occur. It is composed of keratinocytes, melanocytes (located in the SB layer), Merck cells (involved in the touch response, located in the basal layer), and Langerhans cells (involved in the immune protection, located in the SS layer). Underneath the viable epidermis is the

dermis region, which is composed of fibroblasts, dermal dendritic cells, and mast cells [21]. Compared to the avascular epidermis, the dermis and subcutaneous layers contain numerous blood and lymph vessels, nerves, and skin appendages. The subcutaneous layer is the deepest layer of skin directly under the dermal layer, containing half of the body fats. It is responsible for connecting the skin to the fibrous tissue of bones and muscles. Lastly, it is worth mentioning that skin appendages such as hair follicles, sebaceous and sweat glands typically traverse through various skin layers and contribute to various skin functions [21,22]. Figure 2 highlights key anatomical structures of the skin, including the cells present in the various layers.

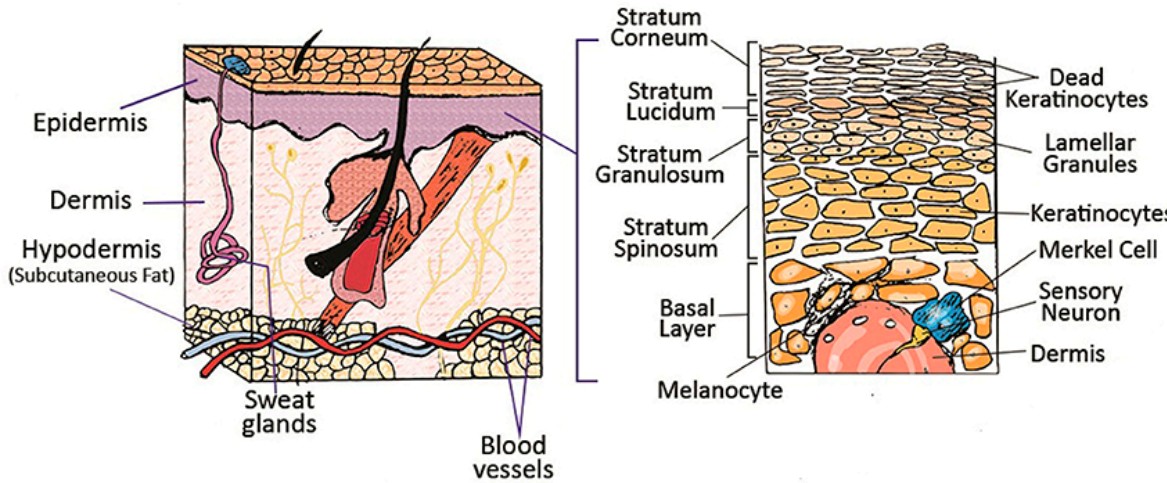

**Figure 2.** Illustration of the skin structure comprising of different layers and appendages. Reproduced with permission from [22]. Copyright © 2016 John Wiley and Sons.

Skin pathology encompasses a diverse range of skin disorders, covering mild to severe conditions, and temporarily to life-long lasting conditions. Atopic dermatitis (AD) and psoriasis are common inflammatory skin conditions occurring primarily in the epidermis layer. Itchy dermatitis and susceptibility to cutaneous infections are characteristic hallmarks of AD. Among other factors, genetics play an important role in the development of AD by inducing defects in the skin barrier function, dry skin condition and immunoglobulin E-mediated sensitization [23]. Family history can be observed among most of the AD patients. Comparatively, psoriasis is an organ-specific autoimmune disease triggered by the altered activation of the cellular immune system. Psoriasis results from the excessive growth and abnormal differentiation of keratinocytes [24,25]. Overall, psoriasis affects roughly 1–3% of the population worldwide [26]. Silver, scaly plaques are the prevalent form of psoriasis (i.e., psoriasis vulgaris), possibly existing anywhere on the skin. However, it is reversible and can be cured with proper treatment. Acne, the most common skin disorder, occurs when hair follicles become clogged by dead skin cells and oil sebum production. Androgen-induced sebum secretion is directly implicated and can indicate the severity of acne symptoms. Famous for its prevalence among teenagers, acne is normally presented on the face, neck, chest, and back area (i.e., up to 20% of teenagers have acne-related facial scars) [27]. Notably, acne is associated with the prevalence of bacterium *Propionibacterium* acnes. Skin infection with human pathogens is another epidemic disease. Suppression of the immune responses is a detrimental virulent strategy adopted by many pathogens, leading to severe infections without proper treatment. As one of the major sources of infection, *Staphylococcus aureus* (*S. aureus*) induce roughly 1.3 million infections per year in US alone [28]. Finally, skin cancer is another type of severe pathological skin condition. Basal cell carcinoma, squamous cell carcinoma (SCC), and melanoma are the main types of skin cancer. Of these, most deaths originate from malignant melanoma located at the basal layer of the epidermis [29]. Some skin conditions (i.e., itchiness/pruritus and dryness/xerosis) are also correlated with systemic organ pathologies, instead of localized skin disorders. For instance, uremic pruritus is found in 40–50% of adult bearing chronic kidney diseases. Calciphylaxis, a life-threatening

vasculopathy of the skin and subcutaneous tissues is reported in 4% of dialysis patients [30]. Meanwhile, diabetes mellitus shows several frequent skin manifestations, including infections (45.7%), xerosis (26.4%), and inflammatory diseases (20.7%) [31].

Efforts to utilize hydrogel technology for various skin pathology are primarily aimed at their treatments, with several recent reviews summarizing these excellent developments [32,33]. Few examples include the development of thermoresponsive composite hydrogel achieving moisture management and drug delivery for atopic dermatitis, thermoresponsive chitosan hydrogel facilitating microenvironment alteration and skin wound healing, as well as injectable micelle/hydrogel composites as curcumin-delivering wound dressing with accelerated healing and antibacterial activity [34–36]. In other studies, cells-laden hydrogels are employed to enable stable engraftment and survival of therapeutic cells (e.g., mesenchymal stem cells/MSCs) to promote vascularization and scar-less wound healing [37,38]. To a lesser extent, hydrogels are also developed to achieve diagnosis and monitoring of these pathological skin conditions. Nevertheless, they are as important as achieving therapeutic purposes. The current article aims to highlight these developments, classified based on the types of analytes for detection: proteins, nucleic acids, bacteria (infection), metabolites, and small molecules, among others. Methodology and techniques employed to fabricate such hydrogels will be described prior to discussing specific study examples. Finally, the remaining challenges and outlook of hydrogel technology for skin applications will be discussed.

## 2. Synthesis and Preparation of Hydrogel Technologies

Hydrogels have been formulated into dressing patches, MNs, as well as microspheres/microparticles to facilitate the diagnosis and monitoring of various skin pathologies. Broadly, the assembly of any hydrogel platform follows two main routes, physical and/or chemical crosslinking. Physically cross-linked gels are formed from physical interactions existing between constituent polymer chains. Such physical assembly offers simple fabrication protocol, and the avoidance of additional cross-linking agents typically required for chemical hydrogel synthesis [39]. Several methods have been identified and employed to achieve physical assembly of hydrogels, including freeze–thawing, through hydrogen bonding or ionic interactions. For instance, poly(vinyl alcohol) (PVA) hydrogels are commonly prepared through repeated freeze–thawing, involving the formation of microcrystals of the polymers and interconnection process through hydrogen bonding [40]. Meanwhile, dispersion of carboxymethyl cellulose (CMC) in HCl solution can lead to replacement of sodium ions in CMC with hydrogen, facilitating the hydrogen-bonded CMC network [41].

Chemically cross-linked hydrogels are fabricated through covalent bonding between the polymer chains, necessitating the reaction of numerous functional groups within the chains. Comparatively, hydrogels assembled in this manner possess stable integrity and strong mechanical properties, allowing for an extended degradation profile under distinct solvents [42]. For example, epichlorohydrin could be used to cross link xanthan and PVA-based hydrogel [43]. Hybrid hydrogel electrodes could be made by dropping hydrogel solutions onto the electrode, in which the polymerization initiator was premixed into the solutions for rapid gelation and uniform film coating on the electrodes [44]. Considering the distinct advantages offered by physical or chemical crosslinking, their utilization could be tailored to suit the chosen gel compositions and intended bioapplications.

Employing the two above-mentioned crosslinking strategies, hydrogel sensors can be conveniently prepared by molding polymer substances together with sensing moieties into desirable morphology (e.g., patches and MNs) or by generating particle droplets through microfluidic technology [45,46]. Typically, hydrogel solution is poured into a metal or polydimethylsiloxane (PDMS)-based template mold, centrifuged to entirely fill the mold spacing (especially for viscous hydrogel solution filling small gaps like MN tips), and crosslinked (e.g., photoinitiator and UV irradiation) to allow hardening and subsequent removal of the gel structures from the mold [17,47]. Colorimetric, enzymatic, and/or electrochemical sensor moieties have been incorporated within hydrogel MNs in this fashion to facilitate the non-invasive assessment and monitoring of skin biomarkers without burdensome devices [48,49].

### 3. Hydrogel-Based Technologies for Point-of-Care (POC) Diagnosis/Monitoring of Skin Pathology

*3.1. Importance of Skin Pathology Diagnosis and Monitoring*

Skin pathology covers a broad spectrum of skin conditions, ranging from a chronic non-closing wound and wound scarring to infections (e.g., acne), gene-related skin diseases (e.g., AD and psoriasis), and skin cancers (e.g., melanoma and squamous cell carcinoma). These pathophysiological conditions can last for a short duration (i.e., several days to few weeks) to those lasting permanently (i.e., a lifelong condition) and affecting a significant population worldwide [50,51]. In USA alone, about 70,000 new cases of melanoma (responsible for most skin cancer-related mortality) are reported yearly, and 1–2% of the population are affected with chronic wounds (i.e., do not close within 12 weeks) throughout their lifetime [52]. Meanwhile, 100 million people in developed countries suffer from abnormal (hypertrophic/keloid) scarring annually [53].

As with other pathological conditions, diagnosis and routine monitoring of skin pathologies impact not only long-term patient outcomes, but potentially also clinical practice as well as economic/insurance policy [54,55]. Timely treatment decision is crucial in ensuring a smooth and efficacious therapy, when pathological burden is relatively low and thorough recovery is highly feasible (e.g., immature scar, which benefits from administration of prophylactics, and management of early-stage skin cancers) [56]. However, this remains a mostly unmet challenge. Period taken from the initial presentation of skin pathology to commencement and re-evaluation of a management plan can be lengthy at times awaiting for discernable visual marks and/or requiring multiple appointments (e.g., clinician, laboratory tests) [55,57]. Such delays and follow-up appointments accumulate and hinder administration of appropriate therapy, leading to the overall increase in cost. In certain cases, tissue biopsy is required for accurate identification of skin diseases, though limited with inconvenience, pain and morbidity of patients, alongside risks of infections and scarring [55].

To this end, point-of care (POC) diagnostic technologies are developed to enable rapid yet sensitive and accurate assessment directly at or nearby the patient care site [57]. Facile POC technologies also promise self-application potential by patients (e.g., self-monitoring of disease progression and assessment of efficacious therapy). Consequently, patients may require less frequent hospital visit for follow-up monitoring, relieving significant medical burden in the long run with minimal or no sensitivity compromised [58].

In recent years, hydrogel applications as POC biosensors have risen significantly, including for several pathological skin conditions. Given its swellable property, hydrogels can facilitate extraction of biofluids (e.g., wound exudate and interstitial fluid/ISF) implicated in various pathological skin conditions [59]. Formed as microparticles or dressing patches, hydrogels enable collection and analysis of open wound exudates [60]. Shaped as MNs, hydrogel MNs enable access through the stratum corneum and epidermis layer for extraction of ISFs, an alternative source of biomarkers especially for localized skin conditions [61]. Relative to blood (conventional source of biomarkers), ISF shares 80–90% of RNA and protein constituents to those found in serum, with comparable relative abundances [62]. Furthermore, ISF is better suited for long-term monitoring with the absence of clotting factors [63]. Following the extraction of skin biomarkers into the hydrogel, detection and analysis can be performed either directly (with sensor moieties pre-embedded inside) or separately in microtubes (facile collection through gentle centrifugation) [64]. For the former, hydrogels mediate contact of the sensor with its target moieties even for small sample volumes, by drawing ISF samples and enhancing receptor/indicator–analyte interactions through space-confinement effects [65]. In the remainder of this section, we will describe in further details the development of these hydrogel technologies for monitoring skin pathology, categorized based on the type of bioanalytes they monitored.

### 3.2. Proteins

Proteins play a crucial role in regulating how cells behave, function, and communicate with other cells and their microenvironment (i.e., through growth factors/GFs and secreted cytokines). Considering its impact in maintaining cellular and tissue physiological functions, it is unsurprising that altered protein expressions reflect most pathophysiological conditions, including for skin [66,67]. For instance, matrix metalloproteinase 2 (MMP2) activity reflects progression of melanoma cancer, alongside the proliferation marker Ki67 [68]. Elevated MMP9 activity is related to the non/poor healing prognosis of chronic skin ulcers [69]. Pathological skin scarring (keloid/hypertrophic scar) overexpresses the transforming growth factor beta (TGF-β) pathway proteins including the connective tissue growth factor (CTGF) and collagen type I, alpha 1 (COL1α1) [70]. T helper type 1 (Th1) cytokines like TNFα, IL-1α, and IL-6 are greatly correlated with psoriasis skin lesions, while Th2 cytokines like IL-4 and IL-13 are indicative of atopic dermatitis [71,72]. Correspondingly, analysis of protein biomarkers in wound exudate as well as ISF is of great importance to monitor local skin pathologies.

In this regard, hydrogel has served as a carrier platform for mediating several sensor moieties, namely sandwich antibodies, cleavable Förster resonance energy transfer (FRET) peptides as well as aptamers. An antibody, with its selective binding ability against certain protein antigens, has conventionally been adapted to the enzyme-linked immunosorbent assay (ELISA) [73]. Incorporated into a hydrogel matrix platform, antibodies enable sensitive detection of proteins through liquid-phase ELISA [74]. For instance, capture antibodies targeting IL-2, IL-4, and TNF-α cytokines can be loaded onto PEGDA gel microparticles through the heterobifunctional PEG linker. In the presence of the target protein, the biotinylated reporter antibody and streptavidin fluorophore will complete the sandwich assay, enabling multiplexed and sensitive detection in pg/mL concentrations [75]. In another example, Zhang et al. demonstrated incorporation of antibody-conjugated photonic crystals (PhCs) possessing unique reflection peaks into PEGDA gel MNs for direct ISF analysis (Figure 3A). Following 40 min of incubation, fluorescent probes could be added to complete the sandwich and enable distinct quantification of each inflammatory marker (i.e., TNF-α, IL-1β, and IL-6) in a sepsis mice model. Crucially, ISF sensing performance was comparable to invasive blood analysis with the standard ELISA technique (Figure 3B) [76].

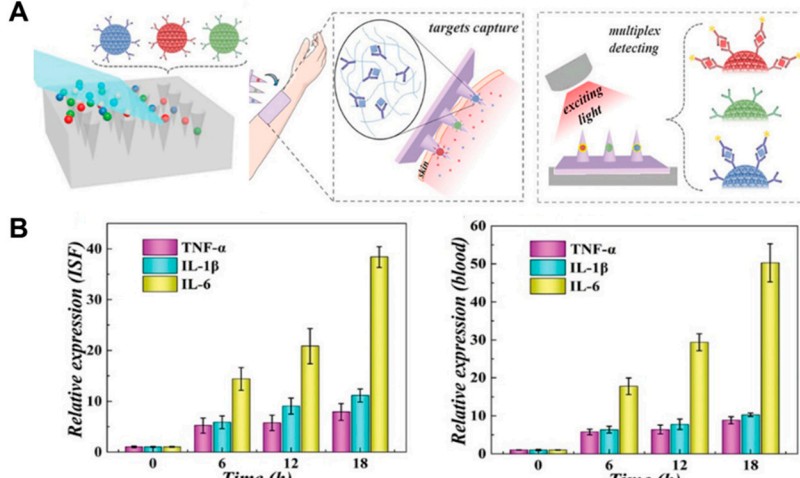

**Figure 3.** Hydrogel-based platforms for monitoring protein biomarkers: (**A**) PEGDA gel microneedles (MNs) loaded with antibody photonic crystal barcodes enable isolation and simultaneous detection of protein biomarkers from interstitial skin fluid. (**B**) Application of the encoded MNs in detecting inflammatory cytokines from sepsis mice (i.e., TNF-α, IL-1β, and IL-6). Detected interstitial fluid (ISF) expressions were greatly comparable to that of the standard ELISA technique of mice blood. Reproduced with permission from [76]. Copyright © 2019 John Wiley and Sons.

As some protein biomarkers function by cleaving other peptides/proteins (i.e., proteinases), FRET peptides that generate a discernable change in the fluorescence signal could report its presence and activity. Kang et al. immobilized such peptides into the PEGDA gel through photo-polymerization to recognize MMP2/9 from the exudate of pressure ulcers as an indicator of its severity. Cleavage occurring between the Gly/Met group separates the Dabcyl quencher from the FITC fluorophore thus restoring its fluorescence [77]. Micropatterned PEG hydrogel modified with MMP9-cleavable FRET peptides was shown to detect subnanomolar MMP9 concentrations in the cell secretome [78]. Comparatively, Shin et al. prepared micrometer-sized PEG gels incorporating the FRET-peptide sensor through thiol-ene photoclick chemistry and embedded it in the secondary PEG/gelatin patch to evaluate collagenase activity of melanoma cells. In this manner, information obtained can have spatiotemporal resolution of tumor progression [79]. Employing similar concept to assemble an activatable chemical probe (i.e., carrying a cleavable peptide substrate and a caged near-infrared dye), we demonstrated sensitive recognition of fibroblasts activation protein α (FAP-α), a crucial biomarker of abnormal skin scarring (e.g., keloid). Coupled with MN platform loading, transdermal delivery of such a probe can facilitate the diagnosis and monitoring of wound scar formation [80].

More recently, aptamer probes capable of binding metabolites, proteins, and surface markers selectively are coupled with hydrogel technology for non-invasive cellular and tissue monitoring, offering clues regarding its status and function [81]. Obtained from an oligonucleotide library through repeated positive and negative selection towards a specific ligand (sequential evolution of ligands by exponential enrichment/SELEX), aptamers are versatile single-stranded RNA/DNA probes undergoing conformational changes upon ligand binding [82]. Microfluidic stop-flow lithography (SFL) was employed to fabricate PEGDA gel microparticles functionalized with aptamer capture sequence for α-thrombin. A pM limit-of-detection (LOD) was achieved with a single capture species [83]. Meanwhile, Tejavibulya et al. immobilized FRET aptamer to PAA hydrogel microfilaments through an acrydite handle for functional ISF sensing. PAA microfilaments possess a high elastic modulus in its dehydrated state to penetrate the skin barrier, and optical transparency in its hydrated state for monitoring of phenylalanine, a crucial biomarker for patients with phenylketonuria [84].

### 3.3. Nucleic Acids

Responsible for directing protein synthesis and thus managing its expression in cellular machinery, nucleic acid (NA) is another type of biomarker widely studied and explored to diagnose and monitor pathological conditions. Among the various kinds of cellular NA, messenger RNA (mRNA; responsible for coding amino acid synthesis and thus protein translation) and micro RNA (miRNA; small non-coding RNA responsible for post-transcriptional gene regulation) are highly correlated to the cellular state and functionality [85]. In several examples involving pathological skin conditions, aberrant WNT5A and MMP1 mRNAs have a detrimental effect in squamous cell carcinoma, while upregulation of mRNAs in the TGFβ pathway (e.g., CTGF mRNA and FAP-α mRNA) are indicative of abnormal wound scarring [86,87]. Krt-16, TNF-α, IFN-γ, IL-23A, IL-12B, CD2, and VEGF mRNA are found to be overexpressed in psoriatic skin lesions [88]. Furthermore, miR-150a and 196 are linked to skin sclerosis, with strong upregulation of miR-92 and 106a regarded as progressive markers for melanoma tumor malignancy [89,90].

mRNAs and miRNAs exist in biofluids including ISF, either as free moieties or within extracellular vesicles (EVs) secreted by cells [91]. Recently, protocols are developed to isolate tissue selective EVs and subsequently RNA moieties from EVs [92]. Accordingly, increasing efforts have tapped on the hydrogel platform to analyze these mRNAs/miRNAs. Facile and non-invasive nature of hydrogel-based isolation makes it highly sought after, as an alternative to biopsy-based isolation, which involves risky surgical procedures and a lengthy recovery, or tape strip-based isolation, which can isolate only limited RNAs from the epidermis region [93].

Given the pairwise hybridization nature of NA (purines-pyrimidines: A-T/U and C-G), it is unsurprising that most NA sensors rely on complementary NAs (natural or

backbone-modified oligonucleotides) tagged with the inducible FRET fluorescence system (i.e., fluorophore-quencher/fluorophore-fluorophore) [94]. Notable examples include the hairpin-shaped molecular beacons, DNAzymes, detachable reporter flares, as well as their formulations on nanoparticles with quenching capacity (e.g., Au nanospheres/nanorods and graphene oxide nanosheets; Figure 4A–C) [95–97]. Interestingly, certain probes such as spherical nucleic acids-based ones are capable of entering exosomal vesicles to bind and directly report the presence of mRNA/miRNA within it [98].

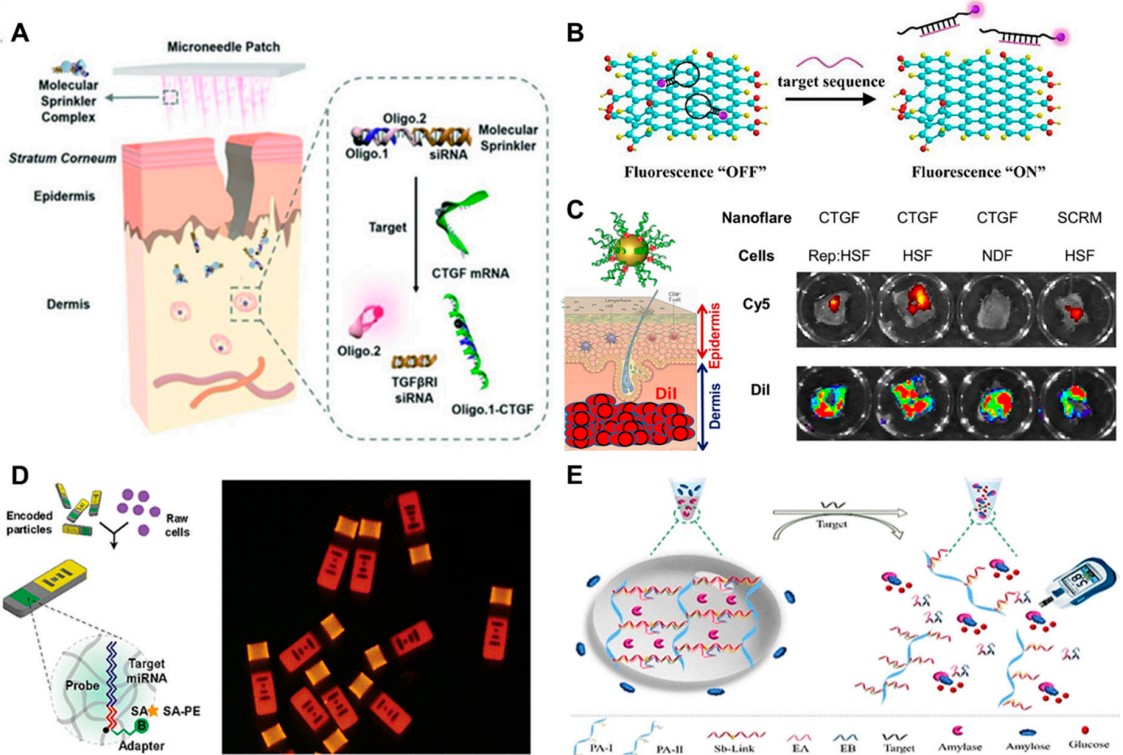

**Figure 4.** Molecular probes, nanotechnology, and hydrogel platforms for detecting nucleic acid biomarkers. Hydrogel MNs or particles can encapsulate or be coated with molecular or nanoprobe assembly for direct/indirect nucleic acid detection: (**A**) Molecular sprinkler assembly for simultaneous detection of CTGF mRNA and delivery of TGFβ-receptor 1 siRNA. Reproduced with permission from [95]. Copyright © 2018 John Wiley and Sons. (**B**) Hairpin-structured probe on graphene oxide nanosheets for recognition of CTGF mRNA. Reproduced with permission from [96]. Copyright © 2018 Elsevier. (**C**) Spherical nucleic acid assembled on Au nanoparticle facilitates topical identification of abnormal scar cells through sensing of CTGF mRNA. Reproduced with permission from [97]. Copyright © 2018 Springer Nature. (**D**) Probe-incorporated PEG microparticles for multiplexed recognition of miRNAs from biological fluids. Reproduced with permission from [99]. Copyright © 2016 American Chemical Society. (**E**) Amylase-trapping DNA-crosslinked hydrogel enables miRNA detection through the glucometer readout following inducible hydrolysis of amylose. Reproduced with permission from [100]. Copyright © 2019 American Chemical Society.

To this end, hydrogel technology for monitoring mRNAs and miRNAs employs two broad detection strategies. In the first method, NA sensor probes are simply embedded into the hydrogel matrix, to capture and eventually present observable signal changes corresponding to the expression level of the diffused RNA sample. Employing PEGDA hydrogel particles with PEG porogen, universal linker ligation, and the biotin/streptavidin labeling strategy, Lee et al. demonstrated versatile detection of miRNA from cells and carrier proteins within 2 h (i.e., miR-21, miR-145, and miR-146a; Figure 4D) [99]. Increasing the size of PEG porogen, the group extended the applicability of the system from small

miRNA to full length mRNA (1000–3700 nt) as shown for beta-2-microglobulin and heat shock protein mRNAs [101]. Adapting a similar PEGDA gel microparticle with distinct code and probe regions (fabricated through stacked microflows) and a universal labeling strategy, Roh et al. demonstrated multiplexed detection of three AD miRNAs [102].

Besides direct incorporation, post-synthesis functionalization of miRNA-specific probes (e.g., through unconverted double bonds of the PEGDA hydrogel) can also be exploited to achieve better loading efficiency and thus detection sensitivity and assay rate [103]. Comparatively, detection sensitivity of the hydrogel sensor can be improved by utilizing the signal amplification strategy, such as rolling circle amplification (RCA). For instance, in the presence of target ACTB and C-myc mRNA within the carboxymethyl HA gel, padlock probes can bind and subsequently be ligated to serve as a template for $\phi$29 DNA polymerase-mediated RCA. Eventually, the 20 nt reporter probe is designed to bind this amplified RCA product, generating a significantly stronger fluorescence than when bonded directly to unamplified target mRNA [104]. Utilizing such a strategy, a limit of detection (LOD) of 20 fM is achieved for multiplexed miRNA-21, miRNA-122, and miRNA-223 detection in the PhC-barcoded PEGDA hydrogel [105].

Alternatively, embedded NA sensors can serve as degradable linkers for the hydrogel structures. In this manner, conformational changes to NA sensors upon target RNA hybridization results in the collapse of the hydrogel structure. Gel degradation can then be monitored and correlated against the RNA expression. Zhu et al. described a colorimetric agent-caging DNA/PAA hydrogel enabling visual target detection. Exploiting aptamers as structural linkers, the group encapsulated AuNP or amylase enzymes within the hydrogel matrix. Upon target recognition, aptamers undergo conformational change, which results in gel degradation and solution color change from AuNP aggregation or amylase reaction with amylose and iodine [106]. Such strategy was later adapted by Si et al. to recognize miRNA by exchanging the aptamer moiety with DNAzyme. In their study, release of entrapped amylase is measured through the glucometer following amylose hydrolysis (Figure 4E), achieving a LOD of 0.3 fM for detection of miRNA-21 among others (miR-335, miR-155, and miR-122) [100]. miRNA-mediated gel degradation can also be measured physically through unblocking of biotin surface-enhanced Raman scattering (SERS) tags from accessing its streptavidin surface sensor [107], or quantifying flow-through distance of the solution in a capillary tube [108].

While most of these examples are not currently targeted at skin pathology, such an adaptation is straightforwardly feasible, through incorporation of these hydrogel sensors into or as a coating layer for the dressing patch or MN platforms. Al Sulaiman et al. described an alginate-peptide nucleic acid-coated MNs for the isolation and detection of NA markers within skin ISF. The platform allows extraction of >6 μL ISF within 2 min and enables marker detection directly on the patch itself or in solution following light-triggered release (degradation of photo-cleavable linkers) with intercalating fluorophores [109]. Bearing in mind the rising importance of miRNA as skin disease biomarkers, it is foreseeable that more hydrogel technology serving such an analysis will be developed in the near future.

### 3.4. Metabolites and Reactive Species

Balance of metabolite production and excretion is crucial for the human health, with significant consequences and illnesses brought by its disturbance. High concentrations of cholesterol and triglycerides in the blood are widely translatable to a high chance of cardiovascular disease [110]. Comparatively, metabolites existing in the ISF or excreted by the skin are highly correlated to pathological skin conditions, and have a great potential as disease biomarkers. For instance, fatty acids like lauric acid and palmitic acid are greatly upregulated in skin melanoma [111]. Lipid mediators that are arachidonic acid metabolites and linoleic acid derivatives are abundant in psoriasis lesions, as compared to non-lesional and healthy skin [112]. Moreover, psoriasis patients are reported to express a higher level of alpha ketoglutaric acid, with downregulation in both asparagine and glutamine [113].

To this end, hydrogel-based micropatches have been developed to isolate and sample skin metabolites for subsequent analysis. Dutkiewicz et al. have invented agarose micropatches for sampling excreted sweat and profiling the metabolite content in combination with direct mass spectrometric screening (Figure 5A) [45]. Such a system could collect and facilitate detection of clinically relevant analytes including lactic acid, serine, threonine, pyroglutamic acid, ornithine, urocanic acid, histidine, and paraxanthine within several minutes. The group then employed a similar setup to evaluate alterations in skin metabolite composition of psoriatic skin. Here, metabolites passively diffused from the skin to the hydrogel matrix with significant differences in several metabolites (i.e., choline, glutamic acid, phenylalanine, lactic acid, urocanic acid, and citrulline) found between the psoriatic and normal skin [114]. Forming a highly swellable methacrylate hyaluronic acid (MeHA) gel MN, our group was able to isolate substantial (>1 μL) ISF within 1 min for subsequent centrifugation retrieval of metabolites and offline ELISA analysis (Figure 5B). Shown for glucose and cholesterol, we obtained great correlation for either marker to the serum measurement [17,61].

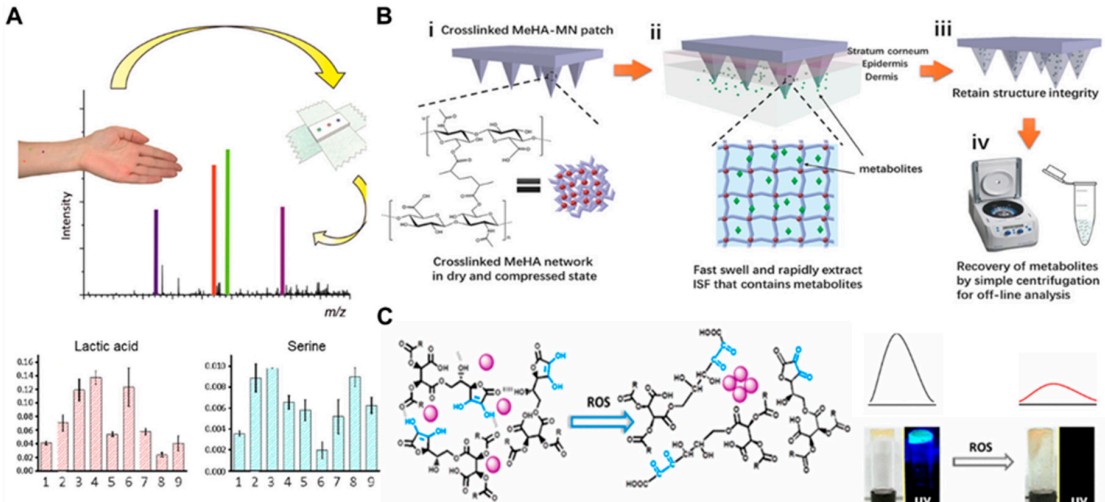

**Figure 5.** Hydrogel-based platforms for the detection of skin metabolites and reactive species. (**A**) Agarose micropatch enables collection of secreted sweat within 10 min for subsequent analysis of skin metabolites (lactic acid, serine, etc.) through electrospray ionization mass spectrometry. Reproduced with permission from [45]. Copyright © 2014 American Chemical Society. (**B**) Cross-linked MeHA MNs facilitates rapid extraction of ISF (>1 μL within 1 min) for offline recovery and analysis of skin metabolites including glucose and cholesterol. Reproduced with permission from [61]. Copyright © 2017 John Wiley and Sons. (**C**) Reactive oxygen species (ROS) sensor based on ascorbic acid hydrogel encapsulating luminescent carbon-dots. ROS-induced oxidation leads to the collapse of the hydrogel framework, aggregation of carbon dots, and quenching of its luminescence. Reproduced with permission from [115]. Copyright © 2016 American Chemical Society.

Alternatively, enzymes-embedded hydrogels are coupled with electrodes or pH-sensitive fluorophores to directly report the presence of metabolites. Li et al. demonstrated a hierarchically nanostructured conducting hydrogel that can detect various metabolites including uric acid, cholesterol, and triglycerides through the current measurement [44]. The Pt NP/polyaniline gel platform are embedded with enzymes that produce $H_2O_2$ and subsequently e- in the presence of the corresponding analyte. Having the features of high substrate permeability and electron transfer, sub-mM detection was achieved rapidly within seconds. A similar concept is also employed for lactate detection, during its oxidation to pyruvate by lactate oxidase (LOx) [116]. Likewise, Bandodkar et al. casted flexible electronics on enzyme-embedded agarose gel to achieve glucose and ethanol detection from sweat excretion. In their study, passive diffusion of metabolites is enhanced through the iontophoresis mechanism to facilitate swift amperometric detection [117,118]. Meanwhile, Srinivasan

et al. incorporated pH-sensitive Coumarin-NH2 into the PEG gel as an indicator for rapid glucose detection. The presence of glucose is sensed by the embedded glucose oxidase enzyme, which produces $H_2O_2$, affects local pH, and correspondingly influences Coumarin-NH2 fluorescence [119].

Reactive oxygen and nitrogen species (ROS/RNS) are additional biomarkers implicated in numerous skin diseases. UV-A, UV-B, and infrared irradiation on the skin results in activation of ROS/RNS signaling, resulting in oxidative degeneration and skin aging [120]. Prolonged ROS signaling during injury aggravates the inflammatory reaction and thus hampers wound resolution [121]. Likewise, superoxide and nitric oxide radicals play a crucial role on psoriasis and AD lesions [71]. Finally, ROS overproduction can stimulate malignant transformation of melanoma [122]. Conventionally, ROS/RNS detection utilizes molecular probes (fluorescent or luminescent), which typically undergo a change in signal yield following chemical reaction (and structural changes) with target ROS/RNS agents [123,124].

Hydrogel-mediated ROS/RNS sensing involves few key strategies. Firstly, optical dyes can be embedded within the hydrogel matrix alongside the ROS/RNS-recognizing elements (e.g., Cytochrome C, Superoxide dismutase, and horseradish peroxide/HRP). For instance, HRP-containing PAA gel spheres report the presence of ROS through diminished dye intensity following HRP-induced oxidation [125]. In another example, ionic ZnS nanoparticles within the microgel structure releases Zn2+ cations in the presence of selective ROS, turning on Zn-responsive Fluozin-3 dye and enabling rapid detection [126]. Considering that ROS/RNS recognition elements typically produces an electron when reacted to its target moiety, immobilization within redox and conducting gel structures can also facilitate rapid and label-free electrochemical quantification. The amplitude of the current generated can then be correlated back to the amount of ROS/RNS in the nearby vicinity [127]. Alternatively, a ROS-induced collapse of the hydrogel framework can be monitored through probe aggregation and subsequent strengthening/quenching of emission spectra. In Figure 5C, Bhattacharya et al. demonstrated an ascorbic acid gel encapsulating luminescent carbon dots for ROS detection. In the presence of ROS moieties, oxidation of the gel framework leads to aggregation of carbon dots and hence quenched luminescence [115].

### 3.5. Bacterial Infections

Skin is a complicated ecosystem resided by various microorganisms. The homeostasis of skin microbiota is important to prevent the invasion of pathogens and keep the skin healthy [128]. Skin pathogens are typically found living on the skin as commensals, but they can be activated due to microbial dysbiosis, host genetic variation, and immune status changes [129]. The detection/monitoring of skin pathogen(s) is important to assess the skin condition, especially during the wound healing process. The gold standard of bacterial detection employs culturing and counting of colony forming units (CFU). Such bacterial isolation and subsequent culture require 2–4 days in general to complete, yet not all bacterial strains could be suitably cultured in the laboratory [130]. Recent development of fluorogenic or chromogenic-based probes to monitor the presence of bacteria-specific enzymes are aimed at reducing resources associated with pathogen detection (i.e., time and costs) [131].

To this end, hydrogels that are widely applied as wound dressing material to stimulate the healing process provide an excellent probe loading site for in situ monitoring of a bacterial skin/wound infection. Ebrahimi et al. recently demonstrated the grafting of the fluorogenic substrate 4-methylumbelliferyl α-ᴅ-glucopyranoside (MUD) and chromogenic substrate N-succinyl-tri-ʟ-alanine 4-nitroanilide (NSAPN) onto chitosan hydrogel films for simultaneous detection of *Staphylococcus aureus* and *Pseudomonas aeruginosa*. The group achieved a limit-of-detection (LOD) of <45 nM and <20 nM within 1 h on their respective enzyme secretion (i.e., α-glucosidase for *S. aureus* and elastase for *P. aeruginosa*; Figure 6A) [132]. In another work, the group employs a similar strategy to detect the enzyme β-glucuronidase (β-GUS), which is secreted by >98% of all known *Escherichia coli* strains. Here, a more impressive detection performance was achieved, with LOD <1 nM within 15 min [133]. Thet et al. designed an agarose hydrogel dressing that can report the presence of the pathogenic wound biofilm. Incorporating degradable lipid vesicles with self-quenching 5, 6-carboxyfluorescein dyes, the group showcased a responsive system in which fluorescence recovery occurred selectively

upon contacting cytotoxic virulence factors secreted by wound pathogens. This system achieves selectivity to pathogenic strains such as *S. aureus* and *P. aeruginosa*, whereas no fluorescent response was observed towards the buffer control and nonpathogenic *E. coli* (Figure 6B) [134]. Considering the plethora of enzymes secreted by different types of bacteria, hydrogel systems can also be designed to undergo enzyme-dependent degradation. Subsequently, the change in the signal expression of loaded probes can be correlated with the presence of target bacteria. For instance, Bhattacharya et al. loaded fluorescent carbon dots within a hydrogel prepared from 6-O-(O-O′-dilauroyltartaryl)-d-glucose, for an esterase-based gel cleavage and detection of several bacteria like *Bacillus cereus*, *Staphylococcus aureus*, and *Bacillus subtilis*. Here, the bacterially secreted esterase results in the fluidization of the hydrogel, aggregation of the embedded carbon dots, and consequently their discernable quenching [135].

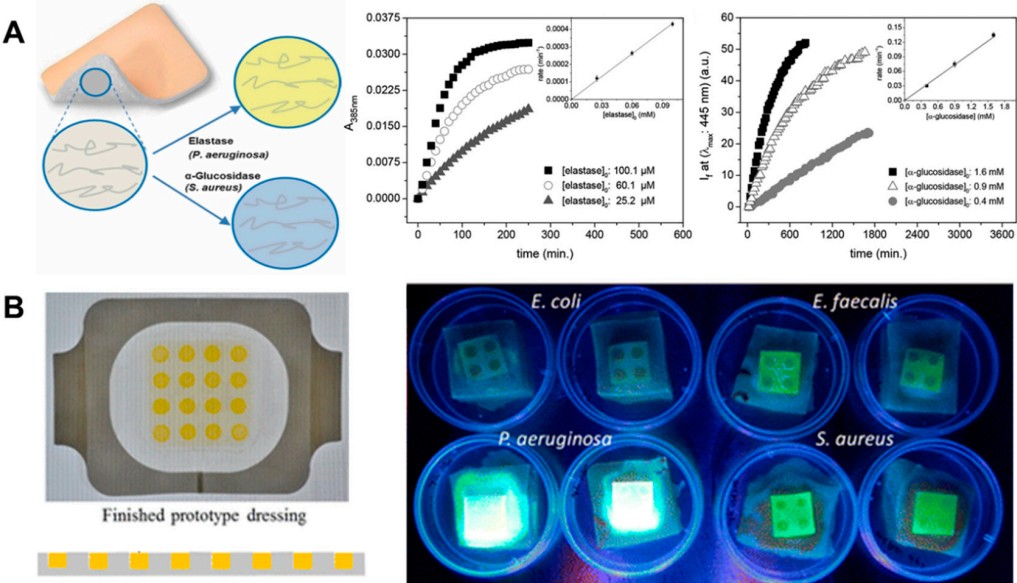

**Figure 6.** Hydrogels for in situ fluorogenic and chromogenic detection of a bacterial infection. (**A**) Chitosan hydrogel films carrying fluorogenic and chromogenic substrate for simultaneous identification of *Pseudomonas aeruginosa* and *Staphylococcus aureus* through their characteristic enzyme production. Reproduced with permission from [132]. Copyright © 2015 John Wiley and Sons. (**B**) Agarose wound dressing embedded with degradable dye-containing vesicles enables selective detection of pathogenic strains. Reproduced with permission from [134]. Copyright © 2016 American Chemical Society.

Alternatively, hydrogel systems can facilitate detection through direct capture of the whole bacteria. For instance, Xu et al. decorated PEG hydrogel inverse opal particles having a characteristic reflection peak with magnetic nanoparticles and aptamer moieties, to achieve specific and sensitive bacteria capture and isolation under controlled magnetic fields [136]. Within 2.5 h, they could identify bacteria at a low concentration of 100 CFU/mL. Meanwhile, Massad-Ivanir et al. developed a porous $SiO_2$/PAA hydrogel hybrid nanostructure as an optical transducer element with monoclonal IgG antibodies as bacteria capture moieties [137]. Upon specific bacteria binding, changes in the optical interference spectrum of the thin film hybrid can be monitored through reflective interferometric Fourier transform spectroscopy (RIFTS). The applicability of such a platform was demonstrated with the detection of low bacterial concentrations from 1000 CFU/mL within minutes. Besides cytotoxins and the direct capture of bacteria, monitoring of bacterial infections can be performed indirectly by measuring skin pH variation. While this will be discussed further in the following subsection, an example of such a strategy was presented by Zepon et al [138]. In their study, a pH-responsive κ–carrageenan/locust bean gum/cranberry hydrogel film (κC:LBG:CB) facilitates a colorimetric detection of bacteria. Relative to its original opaque red color when no bacteria are present (i.e., uninoculated agar surface, pH 7.0), the hydrogel film changes color to dark brownish following contact with inoculated *S. aureus* and *P. aeruginosa* (pH 9.0).

*3.6. Other Markers*

Besides the biological markers described above, there are other general factors that impact skin physiology (and pathophysiology) and provide a predictive role in monitoring their onset and progressions. These include indicators such as pH, temperature, moisture, and pressure. The pH of the skin is slightly acidic, ranging between 4.0 and 6.0. Such an acidic environment is key to support cell proliferation, promote angiogenesis, epithelialization, and thus homeostasis of barrier integrity [139]. Conversely, elevated skin pH is typically associated with pathogen colonization, which utilizes basic pH for enhanced bacterial growth. Correspondingly, the pH of the wound bed is an important marker for wound infection, alongside the overall healing process [140]. To this end, pH-sensing hydrogels have been proposed by incorporating pH-responsive dyes or measuring change in the impedance across composited or polyelectrolyte hydrogels. Tamayol et al. encapsulated pH dye within Si NPs prior to alginate gel incorporation through microfluidic fiber spinning, facilitating a colorimetric pH mapping of the underlying wound (yellow at pH 6 to red at pH 8; Figure 7A) [141]. The color-changing wound dressing was prepared from alginate hydrogel by incorporating pH dyes within resin beads. Subsequently, smartphone imaging was employed to monitor the progression and trigger the release of gentamicin remotely as necessary [60]. Meanwhile, the piezoresistive pH sensor was demonstrated in which pH-dependent swelling of the PVA/PAA hydrogel was used to deflect the silicon base material. Nice correlation was observed between the generated output voltage and the pH of solution measured [142]. Polyelectrolyte hydrogels (composed of ionizable, weak acid and basic moieties) can undergo phase transition near its acid dissociation constant (pKa), resulting in a pH-sensitive change in electrical conductance and hence quantifiable output voltage [143]. Such output voltage can then drive the integrated heater system to induce drug release from embedded nanoparticles and establish an automated theranostic hydrogel dressing [144].

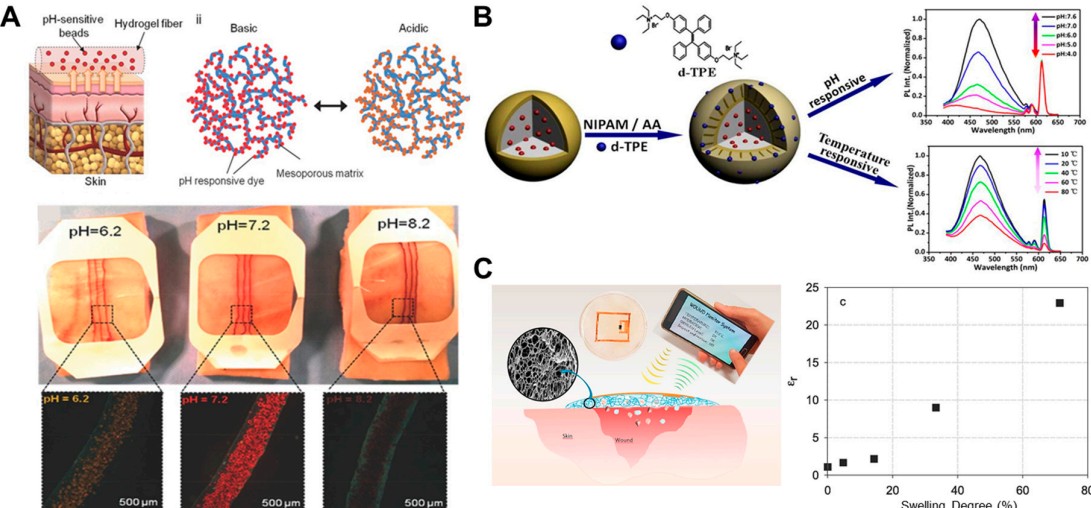

**Figure 7.** Hydrogel monitoring of pathological skin conditions through general indicators including pH, temperature, and moisture. (**A**) pH-responsive hydrogel fibers enable long-term monitoring and an on-site readout of the epidermal wound condition, through incorporation of mesoporous nanoparticles containing the pH-sensitive dye. Reproduced with permission from [141]. Copyright © 2016 John Wiley and Sons. (**B**) Core–shell hydrogel particles consisting of pH- and temperature-responsive copolymers possess dual emission photoluminescence for simultaneous pH and temperature monitoring. Reproduced with permission from [145]. Copyright © 2016 American Chemical Society. (**C**) Hydrogel dressing incorporating the radiofrequency identification (RFID) sensor enables moisture monitoring through swelling-related change in electrical conductivity and permittivity. Reproduced with permission from [146]. Copyright © 2018 Elsevier.

Skin temperature is another general marker possessing great significance in skin pathophysiology. Skin temperature is impacted by local blood flow, extravasation of immune cells (e.g., lymphocytes), wound infection (i.e., extent of colonization) as well as its chronicity [147,148]. In developing hydrogel-based temperature sensors, thermoresponsive hydrogels can be coupled with one or more dyes that undergo an emission shift or change in intensity depending on the swellable hydrogel volume. For instance, the spectral shift of cadmium selenide (CdS)/zinc sulfide quantum dots within PEG hydrogel is correlated to temperatures between 25 and 50 °C [149]. Dual dye-loaded PNIPAM and PAA core/shell gel structures showed temperature-sensitive change in red emission intensity (core) from 10 to 80 °C, while its blue emission relates to varying pH between 6.5 and 7.6 (Figure 7B). Ge et al. [145] reported a polyaniline nanofiber/glycerol/PAA hydrogel sensor capable of distinguishing a change in temperature with the 2.7 °C resolution, based on altered electrical resistance during hydrogel stretching [150].

Moisture is another marker of skin pathophysiology. Several skin conditions, like aging and atopic eczema, results in the loss of skin moisture and the formation of dry and scaled skin [151]. Conversely, chronic and inflamed skin tends to produce significantly more exudate, with enhanced capillary permeability and blood vessel leakiness. A moist environment is crucial in promoting the remodeling stage of wound healing alongside revascularization [152]. However, excessive skin moisture may increase risks of bacterial infection and colonization [153]. Facile monitoring of skin moisture can be accomplished by evaluating the extent by which hydrogel swells following absorption of the skin/wound exudate, and their corresponding change in the dielectric constant (i.e., hydrogel conductivity). In one example, Ajovalasit et al. incorporated an integrated radiofrequency identification (RFID) tag into the PVA/xyloglucan (XG) composite hydrogel. Swelling behavior of the gel affects the RFID tag response, enabling a remote observation (Figure 7C) [146]. Further equipping the chip with an integrated temperature sensor, the RFID epidermal sensor can also facilitate temperature monitoring [154].

Lastly, pressure (mechanical stress) is an important aspect for monitoring pertaining to chronic wound and ulcers. A long term, localized pressure application can induce the formation and development of pressure ulcers, such as in patients undergoing lengthy surgery, in an intensive care unit or those with limited mobility [155]. The development of a pressure sensor can potentially prevent such occurrences while alleviating the healthcare burden of labor-intensive vigilance [156]. Several hydrogel-based pressure sensors employ a similar strategy to what was proposed above, relying on the change of hydrogel conductance or capacitance following its conformational changes. For example, liquid metal particles are included within PVA gel networks for conductivity, permitting the formation of wearable, self-healing, and dissolvable pressure sensor [157]. In another example, ionic hydrogel transduction strategy is utilized as a capacitive pressure sensor with 1 kPa sensitivity, through incorporation of calcium carbonate NPs within the PAA and alginate network [158].

## 4. Conclusions and Outlook

Hydrogel platforms with its unique material properties (i.e., swellable, controlled degradability, and self-healing) are widely sought and developed to achieve various biomedical applications. One key application area includes achieving diagnosis and monitoring of pathological diseases. Hydrogel sensors to assess skin pathology is particularly interesting, offering a safe and user-friendly alternative to tissue biopsy and blood sampling while permitting accurate, real-time, and quantifiable detection [70]. As highlighted through this article, numerous fabrication techniques and types of hydrogel sensors have been elaborated over the past decade or so. Smart formulation design and incorporation of various sensor moieties (e.g., antibodies, molecular beacons, aptamers, and chemical probes) have permitted sensing of a wide range of biomarkers, including but not limited to proteins, nucleic acids, bacteria, small molecules, metabolites, and ions [159]. Notwithstanding the impressive progression of hydrogel technology for monitoring of skin pathology, there remain several challenges preventing its clinical translation and wide utilization. In this section, these challenges will be elucidated, alongside

potential strategies to mitigate and address them. Briefly, they are associated with translating hydrogel sensor performance on human skin tissue, scaling up fabrication volume, as well as ensuring proper and safe clinical usage. A summary of these challenges can be seen in Table 1 below.

**Table 1.** Challenges and potential solutions in translating hydrogel technology to monitor pathological skin conditions.

| Challenges | Descriptions | Potential Solutions |
|---|---|---|
| Sensor sensitivity and consistency | Isolation of sufficient samples: ISF, exudate<br>Complexity with human subjects: complicating diseases and behaviors | Adjusting hydrogel patch/MN, coupling of external technology (e.g., suction force)<br>Multiplexing markers with internal references |
| Scaling-up fabrication of hydrogel technology | Batch-to-batch variation<br>Cost of production<br>Sterilization procedures<br>Prolonging shelf-life | Improvement of fabrication techniques (e.g., soft/photo lithography, 3D printing)<br>Standardized good manufacturing practice<br>Post-synthesis steam sterilization and gamma irradiation<br>Controlled cold and humid storage conditions |
| Proper and safe clinical adaptation | Proper interpretation<br>Risks of inflammation (i.e., erythema and swelling) | Thorough assessment and validation (especially during early stage)<br>Optimization of the application procedures (pre-treatment, timing, frequency, and removal/cleaning) |

Firstly, as with other technological platforms, translating a hydrogel sensor from preclinical animal studies to a complex human subject is met with challenges to maintain sensor performance (i.e., specificity and consistency). Isolation of sufficient fluid samples (e.g., ISF) are necessary despite the thicker epidermal barrier of human skin [61]. For this, hydrogel technology can be coupled with other techniques such as osmotic-driven suction and/or negative pressure application [160]. Moreover, while animal testing typically involves the single disease model, real human diagnosis is often implicated with coexisting diseases and/or widely diverse human behaviors. Skin properties of individual patient can somewhat differ from other patients bearing the same disease [59]. To this end, biofabrication (i.e., 3D bioprinting and electrospinning) of in vitro skin models employing patient-isolated cells provides a better avenue to recapitulate the physio-/pathophysiology of human skin in a personalized manner [161]. Coupled with perfusable vasculature and inclusion of skin appendages and immune-competent cells, these skin constructs allow reliable and robust skin disease modeling (e.g., AD) [162,163]. Accordingly, the functionality and performance of hydrogel sensors in varied skin conditions can be studied more thoroughly without requiring additional animal sacrifices. Concurrently, hydrogel sensors can be designed to achieve multiplexed monitoring of several biomarkers, to facilitate more accurate disease differentiation. Meanwhile, inclusion and normalization with internal reference may reduce bias between patient subjects (e.g., signal from healthy region and longitudinal expression tracking) [164].

Secondly, scaling up the fabrication of hydrogel technology would require careful consideration for the long-term and sustainable business industry. This includes several aspects of hydrogel fabrication, including batch-to-batch variation, shelf-life/storage conditions, sterilization, as well as costs of materials and processes [165,166]. Spatial gel inhomogeneity (with uneven cross-linking density) can influence sensor incorporation and hamper monitoring performance [167]. Nevertheless, constant improvement of fabrication techniques (e.g., involvement of soft/photo-lithography and 3D printing) coupled with standardized, well-defined good manufacturing practices promise reliable hydrogel technologies with good resolution and minimal product deviations [168,169]. In terms of sterility, presterilization of polymers and sensor components alongside assembly under sterile facility ensure great consistency. However, it can be very costly thus mitigating its wide utilization. Conventional UV or heat treatments are also hindered by the sensitive nature of hydrogel technology (i.e., inducing degradation and impacting gelation behaviors) [170]. Fortunately, recent studies have demonstrated that steam-sterilization and gamma-irradiation minimally impact mechanical properties

of HA, poloxamer, and gelatin-based hydrogels [171,172]. Moreover, hydrogel shelf-life can be prolonged (i.e., to months post-assembly) by maintaining them in a cold and humid condition (e.g., 5 °C at 60% relative humidity) [171].

Last but just as important is establishing proper and safe end-user utilization. As mentioned above, signal expression of sensor moieties can vary between individuals and/or application sites. Thus, interpreting, and correlating the biomarker sensor signal with disease progression and/or therapeutic efficacy need to be done cautiously [57]. While the sensor signal can eventually guide clinicians in deciding the route of treatments (catered towards personalized medicine), repeated observation and a thorough signal validation/feedback system would be necessary, particularly during early stage adaptation [55,148]. Additionally, biosafety of the developed hydrogel platform needs to be evaluated rigorously. Risk of skin irritation (e.g., erythema and swelling) is but one aspect to be reviewed on patients bearing various skin conditions [4,173]. Hydrogel application procedures (i.e., pretreatment of skin, application timing, frequency, and removal/cleaning process) must be optimized not only to achieve efficacy, but also guarantee safe, repeated utilization.

Overall, the future looks promising for hydrogel-based monitoring of skin pathologies. While studies involving such technology are predominantly still at the preclinical stage and there is a major gap to wide clinical acceptance, we anticipate a plethora of interesting monitoring applications in the year to come. Automated, self-monitoring and the drug-releasing patch is an intriguing paradigm shift for patients with life-long, recurring skin conditions, one that replaces current passive dressing. Not only will it provide great convenience to patients, it may also relieve significant healthcare burden from routine clinical inspections and follow-up [148]. Noting that skin disorder manifestations are closely linked with several systemic diseases (e.g., cutaneous infections and xerosis in diabetes, pruritus and calciphylaxis in chronic kidney diseases, and cholestatic liver diseases), clinicians can potentially identify and monitor the progression of these chronic diseases through facile hydrogel sensors as well, extending its applicability beyond skin pathologies [174]. Furthermore, through modulation of its mechanical property (i.e., incorporating shear- and heat-responsive moieties), aforementioned hydrogel sensors can be turned injectable/implantable, enabling localized monitoring of particular tissue/organ or systemic evaluation within the blood vessels. Oxygen-reporting hydrogels are important for reporting hypoxic conditions in exercise physiology, tumor therapy, and identification of stroke and ischemic renal injury [175]. Meanwhile, coating of bioimplants with the pH-sensing hydrogel can enable advance warning of bacteria colonization and biofilm formation (e.g., urinary catheters, heart stents, joint implants, and skin grafts) [176,177]. Finally, we hope that this review article will inspire more researchers to carry out innovative and translational research to implement hydrogel technology for diverse biosensing applications.

**Author Contributions:** Conceptualization, C.W. and C.X.; writing—original draft preparation, C.W., X.N., M.C.; writing—review and editing, C.W. and C.X.; funding acquisition, C.X. All authors have read and agreed to the published version of the manuscript.

**Funding:** This research was funded by Singapore Agency for Science, Technology and Research (A*STAR) Science and Engineering Research Council Additive Manufacturing for Biological Materials (AMBM) program (grant number A18A8b0059); internal grant from City University of Hong Kong (grant number #9610472); and General Research Fund (GRF) from University Grant Committee of Hong Kong (UGC) Research Grant Council (RGC) (grant number #9042951).

**Conflicts of Interest:** The authors declare no conflict of interest.

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
