# Peer review of "Hydrogel-Based Technologies for the Diagnosis of Skin Pathology"

_technologies, doi:10.3390/technologies8030047_

Round 1

Reviewer 1 Report

In this manuscript, the authors have given a thorough and comprehensive review on the hydrogel technology in the diagnosis and monitoring of skin pathology. At the meantime, the authors have given a tutorial introduction of hydrogel fabrication, skin pathology, important bio-analytes for monitoring and the state of the art technologies. The publication of this manuscript will be of great value for other researchers in this field. Thus, I strongly recommend this work to be published in "Technologies" in present form.

One comment I would like to make is that although I know the authors focused on the hydrogel application on diagnosis of skin pathology for a good reason, with human body as an integrated whole, some skin diseases may have root cause of other parts, and the hydrogel sensors especially the implanted ones could get access to the chemistry of whole body through skin, it will be interesting and attractive to more readers to have some introductions and discussions of the correlations.  

Author Response

Response: We thank the reviewer for the positive valuation on the comprehensiveness and suitability of our hydrogel review for “Technologies”. We also appreciate the constructive input the reviewer has provided pertaining the correlation other diseases have on pathological skin conditions. Indeed, diseases which impact systemic inflammatory response also present significant skin manifestations. Correspondingly, hydrogel sensors (particularly injected/implanted ones) could be adopted to monitor their progression. Accordingly, we have amended our introduction and our conclusion & outlook section to include these discussions.  

Line 117-122: “Some skin conditions (i.e. itchiness/pruritus, dryness/xerosis) are also correlated with systemic organ pathologies … shows several frequent skin manifestations, including infections (45.7%), xerosis (26.4%), and inflammatory diseases (20.7%).[31]”

Line 644-655: “Noting that skin disorder manifestations are closely linked with several systemic diseases … clinicians can potentially identify and monitor the progression of these chronic diseases through facile hydrogel sensors as well, extending its applicability beyond skin pathologies … bioimplants with pH-sensing hydrogel can enable advance warning of bacteria colonization and biofilm formation (e.g. urinary catheters, heart stents, joint implants, skin grafts).[176,177]”

Reviewer 2 Report

The article present a review about diagnostics based on hydrogel technologies. The paper is well written, compreensible even though I am not a Medical Doctor. The article can be a good starting point for developers of sensors and sensing materials. I recommend this article for publication. 

Author Response

Response: We thank the reviewer for the positive valuation on our review pertaining to hydrogel sensor technology. We are glad to hear that the article can be a good starting ground for other researchers and scientists developing sensor materials, which is actually our main aim in publishing this review article.  

Reviewer 3 Report

This paper is interesting and it focuses on skin diagnosis technologies. However, most of details about the in vitro 3D skin model is still missing. Toxicity testing and skin diseases model are very interesting technology that most of pharmaceutical and cosmetics companies would like to achieve in order to reduce the animal testing steps. It is a big area that should be introduced here as well. Please add more information for this application. Please see a few references below

  1. Randall, M. J., Jüngel, A., Rimann, M., & Wuertz-Kozak, K. (2018). Advances in the Biofabrication of 3D Skin in vitro: Healthy and Pathological Models. Frontiers in bioengineering and biotechnology, 6, 154.
  2. Liu, X., Michael, S., Bharti, K., Ferrer, M., & Song, M. J. (2020). A biofabricated vascularized skin model of atopic dermatitis for preclinical studies. Biofabrication, 12(3), 035002.
  3. Kim, Byoung Soo, et al. "3D cell printing of perfusable vascularized human skin equivalent composed of epidermis, dermis, and hypodermis for better structural recapitulation of native skin." Advanced healthcare materials 8.7 (2019): 1801019.

Author Response

Response: We thank the reviewer for the constructive input to introduce and include more information pertaining to in vitro 3D skin models. Indeed, the development of in vitro skin models have brought an interesting platform for toxicity testing and disease modelling, enabling personalization catered to individual patients and minimizing animal testing requirement. Accordingly, we have included in our discussion section the fabrication and usefulness of these skin models to complement hydrogel sensors in achieving facile and thorough screening.

Line 601-607: “biofabrication of in vitro skin models employing patient-isolated cells provides a better avenue to recapitulate the physio-/pathophysiology of human skin in a personalized manner … varied skin conditions can be studied more thoroughly without requiring additional animal sacrifices.”
